# Immunological and Metabolic Causes of Infertility in Polycystic Ovary Syndrome

**DOI:** 10.3390/biomedicines11061567

**Published:** 2023-05-28

**Authors:** Aleksandra Maria Kicińska, Radoslaw B. Maksym, Magdalena A. Zabielska-Kaczorowska, Aneta Stachowska, Anna Babińska

**Affiliations:** 1Department of Physiology, Faculty of Medicine, Medical University of Gdansk, ul. Debinki 1, 80-210 Gdansk, Poland; a.kicinska@gumed.edu.pl (A.M.K.); magdalena.zabielska@gumed.edu.pl (M.A.Z.-K.);; 21st Department of Obstetrics and Gynecology, Centre for Postgraduate Medical Education, ul. Żelazna 90, 02-004 Warsaw, Poland; radoslaw.maksym@cmkp.edu.pl; 3Department of Biochemistry, Faculty of Medicine, Medical University of Gdansk, ul. Debinki 1, 80-210 Gdansk, Poland; 4Department of Endocrinology and Internal Medicine, Medical University of Gdansk, 80-210 Gdansk, Poland

**Keywords:** low-grade inflammation, immunological factors, metabolic disorders, polycystic ovary syndrome, infertility, progesterone, fertility biomarkers, autoimmunity, adipokines, ovulation and implantation disorders

## Abstract

Infertility has been recognized as a civilizational disease. One of the most common causes of infertility is polycystic ovary syndrome (PCOS). Closely interrelated immunometabolic mechanisms underlie the development of this complex syndrome and lead to infertility. The direct cause of infertility in PCOS is ovulation and implantation disorders caused by low-grade inflammation of ovarian tissue and endometrium which, in turn, result from immune and metabolic system disorders. The systemic immune response, in particular the inflammatory response, in conjunction with metabolic disorders, insulin resistance (IR), hyperadrenalism, insufficient secretion of progesterone, and oxidative stress lead not only to cardiovascular diseases, cancer, autoimmunity, and lipid metabolism disorders but also to infertility. Depending on the genetic and environmental conditions as well as certain cultural factors, some diseases may occur immediately, while others may become apparent years after an infertility diagnosis. Each of them alone can be a significant factor contributing to the development of PCOS and infertility. Further research will allow clinical management protocols to be established for PCOS patients experiencing infertility so that a targeted therapy approach can be applied to the factor underlying and driving the “vicious circle” alongside symptomatic treatment and ovulation stimulation. Hence, therapy of fertility for PCOS should be conducted by interdisciplinary teams of specialists as an in-depth understanding of the molecular relationships and clinical implications between the immunological and metabolic factors that trigger reproductive system disorders is necessary to restore the physiology and homeostasis of the body and, thus, fertility, among PCOS patients.

## 1. Introduction

Infertility is a growing social problem, which is why the World Health Organization (WHO) recognized it as a civilization disease of the 21st century. Although infertility is defined as the inability to conceive despite 12 months of regular sexual intercourse with the intention of having a child in women <35 years of age (American Society for Reproductive Medicine—ASRM), often a term of 6 months of unsuccessful attempts in women >35 years of age is applied in clinical practice within the similar diagnostic management [1]. It is difficult to estimate the sheer scale of the phenomenon and the count of couples of reproductive age in the world struggling with infertility is presented as between 8–12% and even as much as 18%. Most of these are from developing countries [2,3]. The Central and Eastern region, Central and South Asia, and Sub-Saharan Africa have the highest rates of infertility, reaching up to 30% [4].

Infertility is not a uniform entity resulting from a dysfunction exclusive to the reproductive system but a multifactorial one. More and more attention is being paid to the action of the systemic enzymatic response that is dependent on the inflammatory response in connection with functional disorders, insulin resistance (IR), and oxidative stress. They are not only present in cardiovascular diseases but also in infertility [5]. In terms of the conditions and cultural factors, some diseases may occur immediately, while others may not appear until years after the diagnosis of infertility [6]. The syndrome that combines all these health problems and affects women with infertility is polycystic ovary syndrome (PCOS).

## 2. Epidemiology of PCOS

Polycystic ovary syndrome is a common disease with over-representation in infertile women. PCOS contributes to up to 56% of infertility cases [7,8]. The diagnosis criteria for this syndrome are based on the Rotterdam Criteria which were approved in 2003 requiring two out of three criteria to be present: absent or infrequent ovulation, clinically and/or biochemically confirmed hyperandrogenism, and an ultrasound image of polycystic ovaries (the presence of at least 12 follicles with a diameter of 2–9 mm and/or ovarian volume of >10 mL) [1]. Furthermore, in 2006, the Androgen Excess Society recognized hyperandrogenism as a core symptom of PCOS [9]. The current indications for the Rotterdam Criteria are considered the best clinical treatment when it comes to PCOS and the path to obtain the most accurate determination of the individual phenotypes of this syndrome [1].

The results of the analysis of PCOS prevalence in 204 countries in the years 1990–2019 were published in 2022. In 2019, the global point and annual age-standardized PCOS incidence rates increased by almost 30% since 1990 [10]. According to the Global PCOS Treatment Market, Forecast & Opportunities report, PCOS prevalence could reach around 5.1 million by 2025 [11]. It seems that the increase in the incidence of this syndrome is related to the increasingly common diabetes mellitus (DM) and insulin-glucose metabolism disorders, as well as the plague of obesity and oxidative stress associated with a poor lifestyle and excessive supply of simple sugars in the diet [12,13,14,15].

## 3. Etiology of PCOS and Infertility

The etiology of PCOS is multifactorial. The disorders occurring in this endocrinopathy are the result of a complex interaction of genetic, epigenetic, gender, racial, environmental, and stochastic factors. PCOS not only affects the reproductive system but the entire body and accompanies the patient throughout their life [16]. In the youngest women, it is manifested by puberty and menstrual disorders, as well as cosmetic attributes associated with hyperandrogenism. At a later age, problems with ovulation and infertility dominate, followed by metabolic disorders. PCOS is linked to an increased risk of diabetes, dyslipidemia, cardiovascular disease, and fatty liver, as well as a higher risk of cancer, autoimmune diseases, and mental disorders [17,18,19].

One of the links that have been recognized in the pathogenesis of PCOS is insulin resistance (IR), which is induced by inflammation and hormonal dysfunctions originating in adipose tissue. Abnormal glucose and lipid metabolism cause low-grade inflammation in the endothelium. On the other hand, the presence of chronic inflammation may contribute to the development of IR, which is enhanced by adipokines released from adipose tissue [20]. IR also contributes to the excessive secretion of androgens in the absence or reduced production of appropriate concentrations of progesterone, resulting from ovulation disorders. IR-induced disorders are sustained and further aggravated by hormonal and metabolic dysfunctions leading to immune dysregulation. Together, these mechanisms lead to the “vicious circle effect”, giving rise to numerous systemic disorders in PCOS. The result is an abnormal growth of ovarian follicles and disturbed oocyte maturation, as well as the dysfunction of endometrial receptivity, which all contribute to infertility [21]. These disorders and the relationships between them build a kind of pyramid, with infertility being at the very top (Figure 1). This phenomenon can also be depicted as an “iceberg”, the tip of which as a visible symptom above the water’s surface is infertility. The part that is hidden under the ocean’s surface resembles disorders and diseases which, depending on the genetic and environmental conditions as well as the cultural factors related to lifestyle, diet, and exposure to harmful environmental factors, may be revealed immediately or after many years from an infertility diagnosis (Figure 2) [6].

It has long been known that there are genetic factors associated with the inheritance of PCOS (Table 1). However, the high heterogeneity of the described gene sets does not allow for an unambiguous determination of the PCOS genotype [22]. Although the contribution of genes is estimated to be 72%, the genetic loci that have so far been identified as determining the occurrence of this syndrome account for only about 10% [23,24]. Most research suggests that the mechanisms triggering PCOS result from epigenetic changes, including the glycation of the final products of specific genes. These post-translational modifications depend on the environment of the maternal organism, which is increasingly influenced by civilizational factors leading to obesity and hormonal and immunological disorders involved in the pathomechanism of PCOS in the developing fetus (Figure 2) [25]. In addition, there is evidence of a genetically male equivalent of PCOS that rules out the ovary as the starting point for this syndrome [26]. It was shown that men at a high genetic risk of developing the “male counterpart” of PCOS had an increased likelihood of developing obesity, diabetes, cardiovascular disease, and male pattern baldness [27]. It is possible that PCOS-related reproductive dysfunctions may be caused by biological mechanisms that are common to both men and women [26].

## 4. Genetic and Epigenetic Basis for the Development of PCOS and Infertility

It should be emphasized that the pathomechanism of infertility in PCOS begins to be regulated on the gene level (Table 1). Studies have revealed the presence of a family history of PCOS with a genetic background, but the mode of inheritance is not sufficiently clear [28]. In recent years, several studies have described various genes affecting fertility potential that regulate the biochemical pathways, gonadotropin secretion and action, ovarian function, and steroidogenesis.

It has been proven that relatives of women with PCOS, both male and female, are also at increased risk of developing not only reproductive but also metabolic disorders associated with PCOS, such as a higher incidence of insulin resistance, obesity, diabetes, cardiovascular disease, early male pattern baldness, and hypertrichosis [29].

This seems to be an obvious consequence since the abnormality underlying PCOS is indeed an aberration in the gonadotropin/steroidogenesis/insulin pathway associated with inflammation. Thus, ovarian dysfunction in the form of PCOS can only be a consequence of this and one of many further symptoms. Among the studies confirming the influence of hereditary factors on the occurrence of PCOS that merits attention is a Swedish study conducted on a group of over 29,000 women. It was shown that the daughters of women with PCOS had a five-fold increased risk of developing this syndrome than the daughters born to women without PCOS [30]. In a Dutch study on the occurrence of PCOS in identical twins, it was estimated that the heritability of PCOS is about 70% [23].

Although the role of hereditary factors in the occurrence of PCOS and infertility is widely recognized, intensive research is still being conducted on the genetic aspect of infertility in this endocrinopathy. Genome-wide association studies (GWAS) have increasingly identified a number of regions of the human genome that may be responsible for the genetic background of PCOS [31].

In a study on the link between ovarian cancer and PCOS, Zou et al., identified key genes associated with PCOS and ovarian cancer. Using integrated bioinformatics analysis, they found as many as 1061 differentially expressed genes (DEGs) in PCOS patients compared to normal women based on the GSE34526 dataset from the GEO database (GEO Expression Omnibus). In addition, they found 128 common genes with differential expression (DEG) in the progression of PCOS and ovarian cancer [32].

One of the newer large-scale genome-wide association studies (GWAS) linking more than 10,000,000 genetic markers in 10,074 European women with PCOS and 103,164 controls revealed three new loci (near PLGRKT, ZBTB16, and MAPRE1) associated with PCOS risk. The data also provided the first genetic evidence for a male PCOS phenotype associated with male pattern baldness and a causal relationship with depression, which had previously only been reported on the basis of observational studies of these patients. This international meta-analysis of GWAS studies has provided evidence of a common genetic basis for many different symptoms in PCOS, including gender, mental health, and reproductive potential [33].

Steroidogenesis pathways are an important aspect in the etiology of infertility as they are associated with genetic variants that cause clinical hypothalamic-pituitary and gonadotropin receptor dysfunction in PCOS women with infertility problems.

The genes involved in the steroidogenesis process include those from the CYP family: CYP11A, CYP21, CYP17, CYP19, the AR-gene of the androgen receptor as well as the gene encoding sex hormone binding globulin (SHBG) [34].

The disruption of the transcription of these genes leads to hyperandrogenism and ovulation disorders. Among the CYP genes (encoding polypeptides of the cytochrome P450 family), the CYP 19A1 gene is responsible for the activity of aromatase p450, which is also necessary for the formation of estrogens. A defect in this gene causes a decrease in the activity of this enzyme, which leads to hyperandrogenism in both obese and lean women with PCOS. The result of this finding suggests that CYP19A1 contributes to infertility through the adverse effects of androgens on metabolism and inflammation impairing ovulation.

Other genes from the CYP subset are also extremely important in the context of the development of PCOS and infertility. CYP17A1 encodes an enzyme that catalyzes the production of steroidogenesis: androgens, glucocorticoids, and progestins. Mutations in this gene are associated with adrenal hyperplasia, pseudohermaphroditism, and deficiencies of enzymes responsible for changes in the androgen pathway, including progesterone deficiency. In contrast, the CYP11A1 gene encodes a protein found in the inner membrane of mitochondria that catalyzes the conversion of cholesterol to pregnenolone in the first step of steroidogenesis [35].

The previously described metabolic disorders in PCOS that are associated with infertility, such as IR, have also been linked to polymorphisms in specific genes. CAPN10, which encodes calpain 10 and is involved in insulin function and secretion, has been identified as a risk gene for type 2 diabetes, and specific CAPN10 variants have been associated with PCOS [36]. Similarly, specific polymorphisms of the insulin gene and its receptor INSR and the IRS gene encoding the insulin receptor substrate protein, or Fat Mass Obesity (FTO) a gene encoding the alpha-ketoglutarate enzyme associated with obesity and type 2 diabetes, have been associated with disorders of insulin resistance in PCOS [37].

Among the genes that have been discovered and associated with fertility disorders in PCOS, the following should also be mentioned: the follicle-stimulating hormone polypeptide B gene (FSHB), the luteinizing-hormone gene/gonadotropin receptor (LHCGR), the follicle-stimulating hormone receptor gene (FSHR) encoding G-protein coupled receptors essential for the development of the gonads, the androgen receptor gene (AR), the anti-Müllerian hormone (AMH) gene, the DENND1A gene encoding a protein called connecdenn-1 that transports the endosomes contained in the theca cells of the ovary, the overexpression of which is the source of increased androgenization. In addition, SRD5A, a variant of the 5α-reductase type 1 gene responsible for hirsutism, as well as other genes related to metabolism, such as thyroid adenoma-associated (THADA) gene or the already mentioned INSR-insulin receptor gene, associated with insulin resistance PCOS in liver cells, fibroblasts, and skeletal muscle, including also insulin receptor substrate protein (IRS) [37,38].

Among them, the AMH gene deserves special attention. It encodes a protein that is involved in infertility. Whole exome sequencing and GWAS identified rare variants of the AMH gene that have been linked to PCOS. Reduced AMH signaling has been shown to result in a lack of inhibition of CYP17 activity, resulting in increased androgen biosynthesis and impaired fertility [39]. Therefore, high serum AMH levels are not just a result of PCOS but are directly correlated with increased testosterone and LH levels in women with PCOS as well as with altered oocyte maturation and poor embryo quality [40,41]. It should be noted that elevated levels of AMH in the follicular fluid of women with PCOS are correlated with a higher percentage of immature oocytes and lower fertilization rates compared to other infertile women [42].

Subsequent studies have shown that polymorphism in the FSHR gene is significantly associated with PCOS. Two missense mutations were detected, including p.Ala307Thr and p.Asn680Ser. These polymorphisms of the FSHR gene have been shown to have a significant impact on the hormonal response in the ovaries. FSHR mutation rates were also shown to be significantly higher in women with PCOS than in controls [43].

In addition, it has been demonstrated that infertility in women with PCOS may be associated with the occurrence of changes in the HOXA-10 and HOXA-11 genes, which leads to implantation failure as endometrial receptivity is disturbed. HOXA10 promotes cell proliferation during decidualization [44], during which HOXA10 has been shown to be actively involved in promoting cell proliferation by regulating hundreds of genes in transgenic mice [45].

Additionally, of great importance was the discovery that bone morphogenetic protein (BMP), a growth factor in the transforming growth factor-beta (TGF-beta) superfamily, plays an essential role in regulating granulosa cell proliferation and oocyte development. In addition, BMP plays an important role in the process of embryo implantation in the uterine cavity. The loss of BMP signaling in this tissue is due to the BMPRIB mutation and leads to abnormal endometrial gland formation. BMP is essential for estradiol biosynthesis and cumulus cell expansion in vivo. The gene mutation for this protein is associated with a reduced level of aromatase production in granulosa cells, which is why this mutation seems to be one of the key factors responsible for infertility among women with PCOS [43,44,46].

Two polymorphisms of the plasminogen activator inhibitor-1 (PAI-1) gene are also very important in the context of the development of PCOS and infertility: −675 5 G > 4 G (rs1799889) and −844 G > A (rs2227631), which are associated with increased transcriptional activity and PAI-1 protein levels [47]. PAI-1 is an inhibitor of tissue plasminogen and urokinase activators (tPA and uPA), which convert plasminogen to plasmin. PAI-1 is produced primarily in peripheral blood platelets, but it is also produced in placental cells. Increased PAI-1 activity increases the likelihood of clot formation. On the other hand, insufficient activity of this protein increases the risk of bleeding [48].

Additionally, although almost half of the people have the PAI-1 polymorphism in the context of obstetric failures and the coexistence of other unfavorable factors acting synergistically, it is worth evaluating this gene among women with PCOS [49]. Even in a normal pregnancy, the mother’s circulatory system exhibits a physiological state of hypercoagulability due to elevated hormone levels. Therefore, minor changes in the fibrinolytic system of a pregnant woman, such as PAI-1 polymorphism, can lead to hyper- or hypofibrinolysis, affecting the process of placental formation and all the clinical consequences of pregnancy [50].

Complete PAI-1 deficiency is caused by mutations in the SERPINE1 gene, which is located on chromosome 7 (7q21.3-22) and consists of nine exons [51]. Plasminogen activators (PA) play an important role in many stages of reproduction: they are secreted during gametogenesis, and are involved in extracellular proteolytic processes during follicle growth, ovulation, and embryo implantation [52,53].

In the presence of specific polymorphisms of the PAI-1 gene, this protein is overproduced, as a result of which the ovarian plasminogen-plasmin pathway is disturbed. Plasminogen is not converted to plasmin upon follicle rupture and, therefore, does not lead to ovulation in women with PCOS [54]. In addition, the serine protease tPA and the PAI-1 inhibitor have been shown not only to play an important role in the processes of ovulation but also to influence oocyte maturation [55].

One large meta-analysis spanning almost 33 years investigating the role of the PAI-1 gene in adverse pregnancy and gynecological complications described the complex role of the PAI-1 gene in various reproductive failures, including recurrent pregnancy loss (RPL), pre-eclampsia (PE), gestational diabetes (GDM), impaired fetal growth (FGR), repeated implantation failure (RIF), polycystic ovary syndrome (PCOS), and endometriosis [50]. The PAI-1 4G/5G polymorphism has been shown to play a key role in RPL through metabolic, thrombotic, and immunological changes. In addition, PAI-1 may be associated with the occurrence and development of PE and, when overexpressed, leads to excessive fibrin deposition and reduced blood flow in placental circulation, which may result in impaired fetal growth and development (FGR).

Some studies have shown that PAI-1-844G/A polymorphism is associated with an increased risk of developing metabolic syndrome [56]. Therefore, due to the frequent presence of the PAI-1 polymorphism among women with PCOS, they have not only reproductive disorders, but also metabolic disorders, a tendency to obesity, an increased risk of cardiovascular diseases, and venous thromboembolism [57].

Interestingly, the detection of the presence of the PAI-1 polymorphism carries with it important clinical decisions regarding therapy, because overexpression of this protein can be treated, among others, through the use of metformin [50].

PCOS phenotypes are highly heterogeneous, but more and more reports are explaining their origin. PCOS is associated with changes in many molecular factors, not only gene defects but also multiple polymorphisms or nucleotide changes that may interfere with the transcriptional activity of genes [33].

From the studies presented so far, it is clear that some of the genes are overexpressed and manifest clinically, while others are not, depending on the exposure of the body to environmental factors from the moment of conception (Figure 1). Growing evidence suggests that it is the abnormal maternal hormonal environment that contributes to the pathogenesis of PCOS through altered epigenetic programming affecting the developing organism in utero [30]. The background of PCOS development is all the more complicated as more factors appear during the life of the patient that have a key impact on the disclosure and expression of genes related to diabetes, inflammation, cardiovascular disease, and infertility in the granulosa cells of women with PCOS (Figure 3).

On the one hand, it has been proven that the same genes that are involved in the pathomechanism of oxidative stress, lipid metabolism, or insulin signaling are involved in follicle growth arrest and metabolic disorders associated with various PCOS phenotypes. On the other hand, it has been shown that these diverse clinical presentations of PCOS often result not from changes in the DNA sequence itself, but from transgenerational and mitotic hereditary processes, that is, epigenetic changes. Tissue-level epigenetic changes are responsible for changing the phenotype of cells as a response to a changed environment. Chromatin modification, even without additions or deletions to pre-existing DNA, is the trigger for epigenetic reprogramming. Specific mechanisms are involved in this process, such as methylation, hydromethylation, formylation, or carboxylation. DNA methylation is an enzymatic reaction that consists of adding a methyl group, usually at the five carbon of the cytosine pyrimidine ring, and then guanine, which are called dinucleotides. In addition to PCOS, the clinical significance of DNA methylation has been demonstrated in such diseases as cancer, type 2 diabetes, neurodegenerative diseases, and cardiovascular diseases [58].

Women with PCOS have changes in DNA methylation in peripheral and umbilical blood, suggesting a link between the PCOS phenotype and epigenetic changes in systemic and fetal circulation cells (Figure 2). Changes in DNA methylation were also found in the affected tissues, such as the ovary, adipose tissue, and skeletal muscles [9].

Other studies show the observation of various clinical and functional symptoms in PCOS and their occurrence at different ages, depending on the therapeutic interventions, which was associated with phenotypic changes in the affected tissues [59,60].

Knowledge about the possibility of using genetic diagnostics in PCOS may prove very useful in the treatment of infertility in these patients. The ability to influence the modification of DNA methylation, gene expression, and the appropriate phenotype opens up new ways for clinicians to select therapy.

Ovulation disorders are a key mechanism of infertility in PCOS. Restoration of this process is the basic element of infertility treatment among these women, but it turns out that the genetic background is important for the success of this therapy. Treatment with metformin is recommended as a first-line drug to induce ovulation in PCOS patients. Metformin has been shown to regulate DNA methylation by controlling the activity of S-adenosylhomocysteine hydrolase, an S-adenosylhomocysteine hydrolyzing enzyme that inhibits DNMT activity [61]. However, it is not effective in all patients. In one study, as many as 20% of patients failed to achieve the therapeutic goals despite the use of adequate doses of metformin [62]. With the current level of diagnostic capabilities, PCOS patients may potentially benefit from genetic testing using genetic screening tests that will determine the polymorphism in the serine/threonine kinase 11 (STK11) or LKB1 and organic cation transporter 1 (OCT1) genes, respectively affecting the pharmacodynamics and the pharmacokinetics of metformin [63]. The identification of single nucleotide polymorphisms (SNPs) in relevant genes allows the prediction of the effectiveness of therapy and treatment-limiting toxicity to be limited [64]. These polymorphisms may be useful in identifying metformin non-responders who may benefit from alternative treatment.

It should be emphasized, however, that genetic tests are not widely available and their costs often limit the possibility of using them in all women with PCOS and infertility. In addition, to a large extent, the clinical picture and the phenotype of a specific PCOS patient will depend not only on the genetic background but also on the genome modifications resulting from the epigenetic changes occurring throughout her life, starting from the moment of conception (Figure 2 and Figure 3).

## 5. The Role of Adipose Tissue in PCOS-Related Infertility

Many women with PCOS are overweight or obese, depending on the different phenotypes of the syndrome [65]. The presence of excess body fat, even with a normal BMI, plays a key role in causing inflammation in the body. Adipose tissue plays an important role as the largest organ of internal secretion (a.i., converting androgens into estrogens due to the presence of aromatase CYP19A1) not only regulates the body’s energy balance but also affects the immune and reproductive systems and the occurrence of cardiovascular diseases [66,67].

Obesity, especially defined by waist circumference, has been confirmed to have a significant effect on fertility, triggering ovulatory disorders and resulting in androgen excess [68]. Obesity is a metabolic condition characterized by chronic inflammation with high concentrations of pro-inflammatory cytokines, chemokines, and oxidative stress markers in conjunction with macrophage invasion in adipose tissue [69]. In addition, the immune profile of adipose tissue, generating chronic low-grade inflammation, gradually becomes systemic and leads to IR and metabolic diseases [70]. Overweight and obese women experience greater reproductive difficulties regardless of the method of conception or applied fertilization techniques [71]. Fertility disorders are caused by a variety of problems, ranging from hypothalamic-pituitary-ovarian (HPO) axis dysfunction to abnormal oocyte quality and altered endometrial receptivity. The direct impact of obesity on fertility is associated with elevated levels of leptin, triglycerides, and free fatty acids in ovarian follicular fluid, as well as the presence of oxidative stress and inflammatory mediators. Oocyte maturation is disturbed in a lipid-rich environment and in the presence of inflammation, which generates abnormal ovulation and leads to infertility [72,73,74].

An extensive meta-analysis of 30 studies has shown that excess body weight is associated with more severe PCOS symptoms and significantly worsens all metabolic and reproductive outcomes [75].

## 6. Adipokines

Adipose tissue can be considered as an endocrine gland that secretes bioactive peptides (adipokines) that act locally, paracrinely, autocrinely, and systemically through the production of hormones, which may play a key role in infertility with PCOS. Research on adipokines has revealed an important link between the basic factors underlying the pathomechanism of PCOS, such as abdominal obesity, IR, hyperandrogenemia, inflammation, oxidative stress, and reproductive disorders [76,77]. It seems that the modulation of the post-receptor phosphatidylinositol 3-kinase (PI3K) pathway of insulin is one of the basic mechanisms mediating the effect of serum adipokines on the development of PCOS and its consequences [78].

As adipose tissue increases in obesity, the pro- and anti-inflammatory properties of adipokines that are found to be in balance in lean and healthy individuals shift towards pro-inflammatory mediators [79]. Interestingly, adipokines play an important role in integrating systemic metabolism with immune function. Anti-inflammatory adipokines stimulate the expression of anti-inflammatory M2 macrophages and downregulate the expression of pro-inflammatory M1 macrophages. In obesity, adipose tissue-resident macrophages tend to polarize towards M1 macrophages under the influence of cytokines as well as glucose and lipids, producing a pro-inflammatory state in the adipose tissue. They are the primary source of the tumor necrosis factor (TNF), a multifunctional proinflammatory cytokine, and proinflammatory adipokines [80]. Kirichenko et al., in 2022, found that adipokines released by adipocytes or macrophages infiltrating adipose tissue in response to adipose tissue gain induce chronic inflammation, IR, and the development of obesity-related diseases [81]. About 80% of people with PCOS have above-normal BMI values [82]. However, it should be remembered that PCOS also affects women with a normal BMI. There are reports that from 20 to even 50% of women with PCOS are of a normal weight or thin [83].

In one of the latest large-scale reviews of original papers listed in the PubMed database, the activities of individual adipokines were reported to affect the development of PCOS, among others [84].

### 6.1. Chemerin

Another cytokine that is related to PCOS pathogenesis is chemerin. A meta-analysis of a total of 897 participants (524 PCOS patients and 373 healthy patients) confirmed that circulating chemerin concentrations were higher in PCOS patients than in the control group [85]. Chemerin increases insulin resistance in the granulosa cells of the ovarian follicles of PCOS patients [86]. It contributes to the triggering of local ovarian inflammation by recruiting immune cells and binding to blood monocytes expressing chemokine-like receptor 1 (CMKLR1) [87,88]. In a rat model, chemerin has been shown to promote autophagy by inhibiting the phosphoinositide 3-kinase signaling pathway (PI3K/Akt/mTOR) in PCOS [89]. It activates the apoptosis of granulosa cells and inhibits follicle growth, which leads to ovulation disorders and infertility [87]. Chemerin reduces the synthesis of estradiol and progesterone by the cells of growing ovarian follicles and contributes to the development of polycystic ovary morphology [84].

### 6.2. Leptin

In PCOS patients, IR has been shown to increase leptin secretion from white adipose tissue [90]. On the other hand, leptin directly affects the reproductive system by regulating the central secretion of gonadotropin-releasing hormone (GnRH) [91]. As food availability and the body’s energy reserves increase, serum leptin levels rise and stimulate GnRH secretion to promote reproductive function. This results in a disorder of the HPO axis that is typical of PCOS, that is, the excessively frequent stimulation of luteinizing hormone (LH) [92]. In addition, in the reproductive system, leptin initiates the fibrosis and excessive apoptosis of ovarian follicle granulosa cells [84].

Among PCOS patients, the expression of the aromatase gene in granulosa cells is reduced by mitogen-activated protein kinase (MAPK) and PI3K, which is an expression of leptin resistance. Therefore, in patients with PCOS, the concentration of leptin in follicular fluid increases and may cause ovulatory disorders [93]. Kucera et al., showed that in a group of anovulatory women, there were higher concentrations of leptin in follicular fluid, which was considered to be a sensitive marker of fertility disorders associated with anovulation [94]. In addition, high leptin levels may contribute to infertility in women with PCOS by blocking the action of insulin-like growth factor (IGF-1) and impairing follicle-stimulating hormone-dependent (FSH) estradiol synthesis in dominant ovarian follicles [20].

### 6.3. Follistatin and Visfatin

Follistatin and visfatin are two adipokines whose effects on the reproductive system are twofold. Follistatin has anti-angiogenesis properties. It helps to prevent the onset of metabolic diseases by inhibiting the immune system factors from the transforming growth factor (TGF) superfamily. Follistatin contributes to the growth of brown adipose tissue, which prevents obesity and its pro-inflammatory effects in the body [95].

Many studies clearly show that visfatin is elevated in the adipose tissue of obese patients. One meta-analysis of 13 observational studies of visfatin and overweight/obese patients involving 644 subjects showed that plasma visfatin concentrations were higher in persons who were overweight/obese, with DM2, metabolic syndrome, and cardiovascular disease [96]. Numerous studies have shown that the concentration of visfatin among women with PCOS is higher than in women in control groups and that it contributes to IR and metabolic disorders [65,97,98,99,100]. Visfatin increases the production of pro-inflammatory cytokines (IL-6, IL-8, and TNF), as well as transcription factor nuclear factor B (NF-B), activator protein 1 (AP-1), and adhesion molecules [81,101]. Thus, it exerts a pro-inflammatory effect and causes fibrosis associated with IR, fatty liver, increased risk of cardiovascular diseases, and infertility [88,102,103,104].

### 6.4. Adiponectin

The APM1 gene, which encodes adiponectin, is located on chromosome 3q27, close to the site responsible for DM2 and obesity [105]. Hence, it plays an important role in regulating body metabolism. Adiponectin, secreted mainly by white adipose tissue is a very strong insulin-sensitizing agent, increasing the glucose uptake by the skeletal muscles and adipocytes, and stimulates the oxidation fatty acids by decreasing expression CD36 (Cluster of differentiation 36—platelet glycoprotein 4—integral membrane protein, imports fatty acids into cells), and reduces gluconeogenesis in the liver [106]. Adiponectin has also been shown to stimulate plasma lipoprotein lipase, increasing the production of nitrous oxide in endothelial cells, and inducing angiogenesis, thereby mediating both the anti-inflammatory and anti-atherosclerotic effects [107]. Through these mechanisms, it lowers the concentration of glucose, fatty acids, and triglycerides in serum, protecting against the development of DM2 and atherosclerosis [108]. There is a negative correlation between adiponectin concentrations and fasting insulin, glucose, and triglyceride levels [109].

Adiponectin receptors are also found in the central nervous system, as well as on many reproductive system organs: the ovaries, fallopian tube, endometrium, and testes. Human granulosa cells in ovarian follicles have receptors for this protein, which regulate the process of angiogenesis in the ovaries and the synthesis of sex hormones. It has also been demonstrated that adiponectin affects the release of gonadotropin, the normal course of pregnancy, as well as the outcomes of assisted reproduction technologies [78].

Hypoadiponectinemia, through its action reducing IR, atherogenic damage, and inflammation of the vascular endothelium, contributes to disorders leading to PCOS [84]. One study on adiponectin that was conducted in a group of women (as well as men, and in a mouse model) demonstrated that androgens reduce the concentration of adiponectin in plasma and this is associated with a high risk of IR and atherosclerosis in men [109]. Testosterone inhibits the secretion of the adiponectin pool in 3T3-L1 adipocytes, hence, this mechanism seems to be of key importance in the development of IR among women with PCOS [110,111].

The presence of central obesity has been shown to be a risk factor for anovulatory cycles in women with PCOS compared to ovulatory PCOS and corresponding controls despite a similar BMI [112]. The study by Carmina et al. conducted among women with PCOS, showed that adiponectin levels were lower in anovulatory patients with PCOS than in patients with PCOS and regular ovulation provided important insight, which explains the key role that this adipokine has in fertility disorders. The studied PCOS patients who had normal body fat distribution and ovulation did not show IR [113].

It has been shown that adiponectin inhibits the secretion of LH and androgens, thereby promoting ovulation. With its reduced concentrations, there is an increased secretion of LH in the pituitary gland and overproduction of androgens in the theca cells of the ovary, which leads to ovulation disorders. Adiponectin, by activating the PI3K-AKT pathway, reduces IR, hyperandrogenemia, which not only regulates the metabolic system, but also promotes the proper functioning of the ovaries [20,114].

### 6.5. Omentin

Omentin is produced primarily in visceral adipose tissue and is credited with insulin-sensitizing and anti-inflammatory effects. Tang et al., reviewed 13 studies conducted among women with PCOS and showed that circulating levels of omentin-1 are significantly lower in women with PCOS compared to the controls. The findings showed that women with PCOS had elevated levels of interleukin 6 (IL-6), interleukin 18 (IL-18), TNF, retinol-binding protein 4 (RBP-4), resistin, leptin, insulin, LH, testosterone, free testosterone, and Homeostatic Model Assessment (HOMA), but they also had lower omentin-1, ghrelin, and Quantitative Insulin Sensitivity Check Index (QUICKI) values compared to healthy controls [115,116]. Orlik et al., conducted a cross-sectional study among 87 obese women. The study group was divided into two groups of obese women: those with PCOS and those without PCOS. Omentin-1 levels were significantly lower in the PCOS group compared to the non-PCOS group and they were unrelated to body weight. Omentin circulating in the body was positively correlated with serum estradiol and negatively with the LH/FSH ratio, which contributed to ovulation dysfunction in PCOS [117].

Omentin regulation at the ovarian level differs from systemic regulation. In a cohort of women affected by infertility, the expression of chemerin and omentin in follicular fluid and granulosa cells (GC) was selectively increased in both the lean and obese PCOS groups [118]. Similarly, the expression of omentin-1 mRNA in granulosa cells among women with PCOS was almost four times higher than in the controls, regardless of their body mass index (BMI). These findings support the importance of omentin-1 in the pathogenesis of PCOS and suggest that omentin is produced in part by the granulosa cells [64]. Omentin, by acting autocrinely and/or paracrinely on the ovarian follicle cells, modulates their functions, in particular the production of steroids, thereby affecting ovulation in the group of PCOS patients [116]. Thus, central obesity with an unfavorable adipokine profile may trigger an unfavorable hormone-immune profile leading to fertility disorders. However, it should also be borne in mind that there is a large group of women with PCOS and a normal BMI.

Insulin resistance, oxidative stress, and low-grade inflammation and their impact on ovulation disorders

Insulin resistance is one of the main features of PCOS, characterized by a reduced cellular response to the physiological concentrations of insulin, which triggers a state of hyperinsulinemia [119]. The mechanism of IR in PCOS has been attributed to a fundamental defect in the post-receptor insulin pathway of PI3K, which depends on adiponectin and mediates the metabolic effects of insulin. In this mechanism, there is a compensatory increase in insulin released from the pancreatic beta cells, leading to hyperinsulinemia. An alternative post-receptor pathway of insulin, mitogen-activated protein kinase (MAPK), is activated, which results in the release of atherosclerotic mechanisms, excessive steroidogenesis, and mitogenic defects, which may lead to the blast transformation of lymphocytes and, thus, also to carcinogenesis. The deleterious effects of hyperinsulinemia in women with PCOS are pleiotropic and can manifest to varying degrees and at different times in these patients’ lives [100]. Increased plasma concentrations of androgens (both in adulthood and in the prenatal period, depending on the mother’s uterine environment) and increased sensitivity of the androgen receptor are additional factors contributing to IR triggering [120,121].

Women with the conventional PCOS phenotype have been shown to be obese and have higher insulin and IR concentrations [122]. It should be emphasized, however, that thin women with PCOS may also show hyperinsulinemia with IR, mainly due to the accumulation of visceral fat and genetic and metabolic differences [123]. Regardless of the presence of obesity and its obvious effect on insulin sensitivity, the evidence suggests that women with PCOS exhibit an intrinsic form of IR that is unique to this disorder [119].

Although PCOS has been proven to be associated with IR independently of obesity, excess body weight significantly increases both its incidence and severity [100]. In many studies among women with PCOS, higher insulin values than in a healthy control group were confirmed, and the level of insulin was directly related to higher levels of adipose tissue [110,121,124]. Therefore, therapeutic interventions accompanied by an anti-inflammatory diet and a low glycemic index, leading to weight loss and a reduction of visceral fat deposition, play a huge role in infertility therapy. Their implementation leads to the return of ovulatory cycles and improves the pregnancy rate among infertile women with PCOS [125].

Insulin resistance causes hyperglycemia and increased levels of free fatty acids (FFAs). In turn, FFAs, by hyperactivating the electron transport chain, stimulate mononuclear cells to release reactive oxygen species (ROS), which in turn induce oxidative stress [126]. Further, oxidative stress causes cell damage and activates the transcription of pro-inflammatory cytokines such as TNF and IL-6. However, the resulting pro-inflammatory state may contribute to the development of IR and hyperandrogenism, both of which affect ovulation disorders in non-obese women (Figure 4).

It has also been shown that the presence of antioxidants in follicular fluid directly correlates with the success of in vitro fertilization, and superoxide dismutase is involved in the process of embryo implantation in the endometrium. Oxidative stress in the oocyte negatively affects both its meiotic and cytoplasmic maturation, generating oocytes of reduced quality, and inducing abnormalities in the development of the embryo before implantation [127].

Suppression of oxidative stress leads to a reduction in apoptosis and has therapeutic potential in this regard, most likely by protecting mitochondrial function and regulating the proteins associated with programmed cell death [128]. Mitochondria are responsible for the production of the most reactive oxygen species, which induce oxidative stress [129]. Mitochondrial malfunction is responsible for the development of obesity, DM2, atherosclerosis, cardiovascular disease, metabolic syndrome, and cancer, the rate of which also increases in PCOS and contributes to reproductive failure. Moreover, mitochondria are sensitive to environmental changes; therefore, the listed diseases, along with PCOS, have a multifactorial etiology of genetic and epigenetic origin. They reveal themselves depending on the existence of additional factors [130].

Mitochondrial dysfunction in the leukocytes of women with PCOS and IR is manifest in a decrease in mitochondrial oxygen (O^2^) consumption, glutathione, MMP, and mitochondrial membrane potential, impaired oxidative phosphorylation complex (OXPHOS) activity, as well as a significant increase in the production of ROS [131]. Besides, women with PCOS and IR showed impaired endothelial and mitochondrial function in leukocytes, along with an increase in inflammatory markers. The impaired leukocyte-endothelial interaction may be responsible for the increased risk of cardiometabolic diseases in these women [132].

Other studies have shown that the presence of oxidative stress associated with increased production and accumulation of ROS in the female reproductive system negatively affects folliculogenesis and oogenesis. On the one hand, the very presence of ROS is essential for proper folliculogenesis because some of their forms are involved in signaling cascades, such as nuclear factor kappa B (NF-kB) and vascular endothelial growth factor (VEGF), which are necessary for follicle growth. On the other hand, excess ROS can lead to impaired oocyte quality due to mitochondrial dysfunction, decreased ATP production, oocyte aging, and oocyte/embryo aneuploidy [133]. Additionally, excess ROS is generated by elevated androgen levels in women with PCOS due to the high activity of the P450 enzyme. Furthermore, excessive GnRH stimulation, which induces elevated LH levels in women with PCOS, contributes to the disturbed oxidation-reduction balance in the ovary. LH has been shown to play a part in the pre-ovulatory signaling production of ROS by the ovaries by inhibiting hydrogen peroxide (H_2_O_2_) activity. Therefore, in PCOS, increased LH stimulation leads to an increase in the amount of ROS, thus, generating excessive oxidative stress [134]. Turan et al., conducted a study in which they showed a relationship between oxidative stress markers and fertility status in PCOS patients. In the reported results, infertile PCOS patients had significantly higher malondialdehyde (MDA) levels and significantly lower thiol levels compared to fertile PCOS patients. Therefore, it was concluded that lower levels of thiol and higher levels of antioxidant enzymes may indicate a compensatory antioxidant mechanism in the studied young and lean female patients. Thus, infertility and IR were found to be associated with increased oxidative stress in young, non-obese Turkish women with PCOS [135].

In a study of Iranian women with PCOS, it was demonstrated that they had increased oxidative stress, as measured by the Advanced Protein Oxidation Products (AOPP), and reduced antioxidant capacity compared to healthy controls matched for age and BMI. AOPPs were recognized as new markers of oxidant-mediated protein damage and function as a new class of pro-inflammatory mediators. Much higher levels of AOPP and a much lower total antioxidant status may contribute to the increased risk of cardiovascular disease in women with PCOS, in addition to the already known risk factors, such as IR, hypertension, central obesity, and dyslipidemia [136].

One study also showed that the pathophysiology of PCOS is a significant contributor to oxidative stress, independent of IR and BMI. In follicular fluid and granulosa cells, 71 women with PCOS and 50 control women undergoing in vitro fertilization (IVF) were tested for the activity and levels of redox markers and transcript expression, respectively. Superoxide dismutase, glutathione reductase, glutathione peroxidase, and paraoxonase-1 (PON1) activity were significantly lower in the follicular fluid of women with PCOS than in the control group [133]. Özer et al., then discovered that infertile PCOS patients had significantly higher malondialdehyde levels, as well as lower serum catalase and zinc levels than fertile PCOS patients. The study concluded that PCOS patients are exposed to oxidative stress, which seems to be the highest in patients with IR and infertility [137].

The oxidoreductive stress occurring in PCOS should be viewed in many dimensions. It turns out that adipokines also play a significant role here. The protective effect of omentin against oxidative stress was determined by quantifying the reactivity of antioxidant enzymes. In studies by Shi et al., it was shown that omentin can attenuate hydrogen peroxide (H_2_O_2_)-induced cytotoxicity and cell apoptosis by inhibiting the expression of the pro-apoptotic Bax protein [138]. Hence, a low level of this adipokine has an indisputable effect on ovulation disorders. By generating increased oxidative stress, it causes abnormal folliculogenesis and oogenesis, which contribute to infertility in PCOS [115].

## 7. Insulin Resistance, IGF-1, Androgens, and Ovulation Disorders

IR generates hyperandrogenicity, which has been recognized as one of the main features of PCOS. Compensatory hyperinsulinism, through the IGF-1, results in ovarian theca cell hyperactivity, leading to the failure of FSH-dependent aromatase and androgen overproduction [139]. IGF-1, through its receptors located in tissues and reproductive cells, directly regulates the development of gametes and affects their quality. It is also a modulator of the steroidogenesis processes taking place in granulosa cells through signaling pathways: JAK/STAT, PI3K/AKT, and pro-MAPK/ERK1/2, which play an important role in ovarian function and are crucial for oocyte development [140].

In women with PCOS, metabolic disorders and increased oxidative stress lead to the abnormal functioning of the theca and granulosa cells of the ovary through the excessive insulin signaling of the IGF-1 receptors in the ovary. The ovarian reserve, oocyte survival and quality are impaired by these mechanisms [141]. In addition, IR with compensatory hyperinsulinemia induces the overproduction of androgens in the ovaries through IGF-1 receptors, leading to hyperandrogenism and its sequelae [142]. It should also be mentioned that androgenization in PCOS not only has an ovarian but also an adrenal source. Insulin increases basal and adrenocorticotropic hormone (ACTH)-stimulated androgen production and P450c17 expression in cultured human adrenal cells [143,144]. Silva et al., showed in their study that IGF-1 was the only biomarker that reached significantly lower levels in the follicular fluid in the group of women with reduced fertility compared to the control group. Interestingly, a statistically significant difference was confirmed only in the group of women with anovulation. These results support the theory that IGF-1 is an autocrine stimulator of follicular growth [145].

Hyperandrogenism exacerbates both local and systemic inflammation, which can lead to IR. This is another aspect of the interpenetration of the systems and interactions between metabolism, hormones, and cytokines. Hormonal and metabolic disorders trigger immune system dysfunction, which in turn exacerbates metabolic and hormonal dysfunctions, which collectively result in infertility (Figure 4). Androgens stimulate the infiltration of monocytes into the ovarian tissue and increase the secretion of pro-inflammatory cytokines: IL6, TNF, IL1β, IL17, and IL18, which block the maturation of growing follicles. Numerous anovulatory follicles develop and form cystic ovaries in this syndrome, resulting in impaired fertility. However, the process of revealing polycystic ovarian morphology is conditioned by other factors, such as excessive LH stimulation due to a disturbed hypothalamic-pituitary-ovarian (HPO) axis (Figure 3). According to the most commonly used Rotterdam criteria (ESHRE/ASRM, 2004), a clearly defined image of polycystic ovaries on ultrasound is not a necessary clinical feature for the diagnosis of PCOS. However, an increased number of antral follicles, resulting in increased ovarian volume, may effectively hinder the processes of maturation, dominance, and rupture of the ovarian follicle [16].

Furthermore, an excessive surge in LH stimulates the expression of CYP17 mRNA and increases androgen production in theca cells via the PI3K/AKT pathway [146]. Androgens trigger ovarian tissue inflammation and disrupt ovarian follicle maturation by stimulating the secretion of pro-inflammatory cytokines, which leads to ovulation disorders and infertility [147].

There are numerous reports showing a negative relationship between androgen levels and signs of chronic inflammation. Rudnicka et al., studying a group of 200 women with PCOS, showed a close relationship between an increased number of peripheral blood leukocytes and testosterone, androstenedione, and dehydroepiandrosterone sulfate (DHEAS) concentrations [148]. Increased levels of androgens and the presence of inflammation trigger fertility disorders in women with PCOS. Gallinelli et al., demonstrated a direct negative effect of androgenemia on ovulation through the association of high levels of androgens with existing inflammation in the follicular fluid of PCOS patients. Reduced numbers of activated T helper (Th) cells and interleukin-12 (IL-12) concentrations have been reported in the follicular fluid of these women [149].

Androgen excess, one of the key characteristics of women with PCOS, also inhibits glyoxylase I activity [150]. Reducing the activity of this enzyme leads to the accumulation of cytotoxic advanced glycation end products (AGEs) in the body, which are formed in women with PCOS as a result of impaired carbohydrate metabolism. Glyoxylase I plays an important role as an enzymatic system that captures 2-oxaldehydes, the main precursors of AGEs. Advanced glycation end products are highly reactive molecules that result from the non-enzymatic reactions of sugars with proteins, nucleic acids, and lipids. Their accumulation in the ovaries leads to the already discussed oxidative stress and damage to all types of cells, which result in disorders in the functioning of the gonads and infertility among women with PCOS [151].

Numerous reports have also indicated the existence of an opposite mechanism wherein it is possible that AGEs play a key role in enhancing androgen biosynthesis in PCOS patients. This is mediated not only by the previously described direct enzymatic change but also by IR and inflammatory induction. Serum AGE levels in women with PCOS have been shown to be positively correlated with testosterone levels, regardless of BMI [152]. Mouanness et al., in a recent review, reported a number of data regarding the role of AGEs in the key elements of the PCOS phenotype and pathophysiology. There is increasing evidence that women with PCOS have elevated serum AGE levels associated with IR and that AGE deposition in ovarian tissue results in anovulation and hyperandrogenism. The literature to date suggests that the interaction of AGEs with their RAGE membrane receptors activates signaling pathways leading to increased oxidative stress, inflammation, ovulatory dysfunction, hyperandrogenism, IR, and obesity. AGE products have been shown to have a toxic effect on ovarian granulosa cells, especially on cell proliferation and hormone release. Therefore, they unquestionably are an important link when it comes to understanding the causes of fertility problems in women with PCOS [153].

## 8. Dysfunction of the Immune System, Inflammation, and Ovulation Disorders

Analyzing the individual elements of the pathomechanism leading to the formation of PCOS, it can be seen that they ultimately lead to an imbalance in the immune system. In PCOS, the key fact is that ovulation is a condition dependent on the occurrence of local, self-limiting inflammation [154]. The inflammatory response necessary for ovulation to take place occurs in two stages. First, vasodilation occurs with the release of chemokines and cytokines, which leads to the infiltration of the cells of the immune system and the digestion of the follicle wall. The immune cells then trigger the local production of molecular mediators that suppress the inflammatory stimulus to confine the process to the follicle wall and provide the proper environment for the fertilization of the oocyte released from the follicle [155].

This normal, two-stage course of ovulation is disrupted by the pathological, chronic inflammation of the ovary caused by IR and hyperandrogenism [156]. It directly causes a state of infertility and reduced fertility in women with PCOS through disturbed follicle growth and oocyte maturation processes [157]. In the described situation, due to the uncontrolled secretion of pro-inflammatory cytokines within the ovary, there is an additional lack of synchronization in the maturation of the oocyte and the rupture of the wall of the follicle in which it is located [158].

Chronic systemic inflammation has been confirmed to exist among women with PCOS (Figure 3). C-reactive protein (CRP), interleukins IL-1, IL-17, IL-1, TNF, monocyte chemoattractant protein 1 (MCP1), soluble endothelial leukocyte adhesion molecule (sE-selectin), and soluble intercellular adhesion molecule (sICAM) are all examples of inflammatory mediators. In one of the review papers, it was noted that the concentration of CRP was twice as high in women with PCOS than in healthy controls [159].

It turns out that the success of reproduction involving follicle growth, oocyte maturation, and successful ovulation and implantation- is based on processes that are ultimately regulated at the immune system level (Figure 4). Cytokines and chemokines have a critical role during a typical menstrual cycle and affect normal ovarian and endometrial function. They are secreted in granulosa cells by leukocytes that normally reside in or periodically flow into ovarian tissue. It has been shown that there is an imbalance of leukocytes in the theca layer of the ovaries in women suffering from PCOS. Ovarian leukocytes from women with PCOS show a different cytokine profile than leukocytes from women without PCOS. Wu et al. showed that CD45RO+ lymphocytes (activated memory T cells) are reduced by as much as 60% compared to the physiological state, while the relative abundance of macrophages and neutrophils remains unchanged [160]. The study also found a relationship between high BMI and TNF and low expression of IL-6 mRNA in follicular cells. It has been shown that IL-6 and TNF are positively correlated with the concentration of lipids in the follicular fluid of obese women, and IL-6 expression was higher in those who had problems getting pregnant. The ovarian lymphocytes of women with PCOS produced elevated amounts of cytokines by Th1 lymphocytes, IFN-γ, and IL-2 in vitro; the alveolar fluid levels were also elevated. The predominance of Th1 lymphocytes in the ovaries of PCOS women may be related to ovarian mechanisms that cause anovulation [161]. The described compilation of events shows a smooth transition from metabolic disorders to the occurrence of oxidative stress, inflammation, and reduced fertility.

It has been shown that almost 30% of women with PCOS experiencing ovulatory disorders are resistant to stimulation with clomiphene citrate, which induces follicular growth through the action of FSH. Among these patients, there was a positive correlation between the serum levels of the pro-inflammatory chemokine CXCL16 and the inflammatory protein CRP. CXCL-16 may impair the ovarian response to clomiphene citrate by increasing angiogenesis and inflammation [162], thus, blocking ovulation and leading to problems with achieving pregnancy.

The systemic inflammation in PCOS caused by IR and lipotoxicity, which are often associated with obesity or visceral adipose tissue dysfunction, can be transmitted to the ovary and follicular fluid by triggering cytokine imbalances. Optimal cytokine expression is generally required for normal ovarian function. However, increased or abnormal secretion of pro-inflammatory cytokines at inappropriate time points during the cycle impairs the normal course of ovulation and fertility. This is because ovulation is an inflammatory process during which a large pool of leukocytes is recruited to the preovulatory follicle wall [158].

In contrast, systemic, chronic inflammation triggered by PCOS may interfere with the regulation of this locally ongoing inflammatory process during ovulation. In response to the LH signaling from the periovulatory surge by the pituitary gland, granulosa cells begin to secrete steroid hormones and inflammatory mediators necessary to initiate the rupture of the ovulatory follicle wall. They include prostaglandins, chemokines, and cytokines, which also have a chemotactic effect on leukocytes and other cells of the immune system that begin to flow into the ovarian tissue [127]. In the area of the ovarian follicle growing towards ovulation, a very large number of cytokines are produced, which also act as chemokines and regulate follicle growth and the rupture of the wall. We have to take into account granulocyte and macrophage colony-stimulating factor (GM-CSF), monocyte chemotactic protein 1 (MCP-1), and IL-8, which play a key role in the recruitment of leukocytes and macrophages to the ovary [163,164]. It is most likely that granulocyte-macrophage colony-stimulating factor (GM-CSF) and IL-8 are responsible for the beginning of the luteinization of the follicle granulosa cells during the process of ovulation. Other studies have shown that the pro-inflammatory cytokine IL-1 is directly responsible for the induction of ovulation [165]. On the other hand, IL-6, interferon alpha (IFNα) and interferon beta (IFNβ) regulate the expansion of the cumulus oophorus after the LH surge and directly contribute to the ejection of the ovum from the follicle [166,167]. The incoming leukocytes (including lymphocytes, granulocytes, and macrophages) stimulate ovulation by secreting proteases and vasoactive agents that directly digest the follicular and ovarian walls.

Therefore, understanding these subtle mechanisms may contribute to improving the treatment of infertility due to ovulatory disorders in women with PCOS resulting from abnormal leukocyte function. The administration in stimulated cycles of not only the so far widely used HCG, which mimics the pituitary LH surge, but also of GM-CSF, may enhance the recruitment of leukocytes to the follicle wall and the normal wall rupture process. Leukocytes recruited to the periovulatory follicle in response to chemokines secreted by luteinized granulosa cells induce the angiogenesis and luteinization of subsequent granulosa cells, promoting the process of ovulation [168]. Similar protocols have been successfully used in the treatment of luteinized unruptured follicle (LUF) syndrome, wherein an improvement of pregnancy outcomes was achieved by administering bHCG and GM-CSF to women having stimulated cycles several hours before the planned induction of ovulation in the ovarian follicle, which stimulated growth [115].

Qin et al., showed that the ovarian leukocytes of women with PCOS have a different cytokine profile compared to healthy controls [169]. Using flow cytometry and ELISA tests, they discovered that the production of Th1 cytokines (IFN-γ and IL-2) in follicular fluid lymphocytes was significantly higher in PCOS patients than in the control group. Th1 immune dominance seems to be a key feature of the ovary in PCOS patients and undoubtedly participates in the immunological pathogenesis of ovulation disorders. This study once again confirmed that chronic inflammation is one of the mechanisms underlying the pathogenesis of PCOS (Figure 3).

## 9. Progesterone Deficiency in PCOS and Immune Infertility

An extremely important aspect of the infertility etiology of women with PCOS is the low level of progesterone generated by the absence of or rarely occurring ovulation. Progesterone not only ensures the cyclic exfoliation of the endometrium and the occurrence of menstruation but, above all, is an immunomodulatory factor (Figure 4). As it turns out, it affects not only procreation and reproduction but also women’s general health [170]. Progesterone is physiologically produced by the luteinized granulosa cells of the ovulatory follicle, which produce estradiol alone in the first phase of the cycle. Progesterone physiologically slows down the GnRH and LH impulses and contributes to the proper functioning of the HPO axis. It has been shown that with normalized pulses of central GnRH and LH secretion, excess androgens and hyperinsulinemia return to normal, and the physiological balance between estradiol and progesterone in a woman’s monthly cycle is restored [170,171]. There are reports, experiences, and good clinical practice indicating that cyclic progesterone administration in PCOS patients improves reproductive health, pregnancy outcomes, and overall well-being. However, attention should be paid to fertility bioindicators in order to supplement in accordance with the patient’s natural menstrual cycle [172].

PCOS is characterized by the central dysregulation of the HPO axis with a rapid pulsation of GnRH, which is followed by a rapid pulsation of LH. This, in turn, impairs FSH-dependent follicular growth and generates anovulatory cycles with a chronic progesterone deficiency. Subsequently, there is the effect of an overproduction of androgens by the theca cells of the ovary and the failure of FSH-dependent aromatase. There is an increased release of androgens from the ovary into the bloodstream and, consequently, IR and inflammation, as well as clinical symptoms specific to PCOS, such as hirsutism, acne, androgenetic alopecia, and ovulation disorders [173].

In a properly functioning ovary, the periovulatory release of the LH from the pituitary induces a transient increase in the expression of the progesterone receptor in the granulosa cells of the preovulatory follicles. Following the LH action, a portion of the granulosa cells of the ovulating follicle is gradually luteinized, releasing pre-ovulatory progesterone, which has a paracrine effect. It has been shown that progesterone is an important mediator of ovulation as it affects the regulation of genes located in the granulosa cells of ovulatory follicles, for example, those encoding proteases a disintegrin and metalloproteinase with thrombospondin motifs 1 (ADAMST1) and cathepsin L (CTSL). Proteolytic enzymes digest the proteins of the follicle wall and help in its disintegration during ovulation [158]. It has been shown that mice with progesterone receptor knockouts in the granulosa cells of the ovulatory follicle were unable to ovulate, despite the use of high doses of exogenous gonadotrophins [174].

The exact mechanisms by which progesterone regulates ovulation are very complex. Its flagship effect is the effect on the release of cytokines and the regulation of immune system genes that affect the recruitment of neutrophils immediately before ovulation. The influx of neutrophils into the follicle wall is necessary for the further stages of rupture of the ovulatory follicle wall. Another important aspect is the proper activity of prostaglandin 2 (PGE2), which acts as an inflammatory mediator within the follicle, inducing the breakdown of the basement membrane, luteinization of the follicle wall, follicle rupture, and the release of the egg cell from the ovary. The enzyme that controls prostaglandin production is prostaglandin-endoperoxide synthase 2 (PTGS2), which is strongly induced by progesterone in ovarian granulosa cells in response to the LH surge [175].

In addition, in PCOS patients, rare or abnormal ovulation implies an impaired receptivity of the endometrium and reproductive failures in terms of embryo implantation and placental development. It has been repeatedly shown that the immunomodulatory effect of progesterone in the endometrium promotes embryo implantation and pregnancy maintenance [176,177]. Progesterone-induced regulatory T cells (Tregs) exhibit immunosuppressive activity and protect the fetus against immune rejection by the mother. Progesterone-promoted Treg cells have been shown to express CD4, CD25, and FOXP3 (a transcription factor responsible for the formation of regulatory T cells), and inhibit the pro-inflammatory effect of other T cells. Foxp3 deficiency results in generalized autoimmune inflammation. Through these mechanisms, Tregs play a key role in the peripheral tolerance of autoreactive T cells and prevent peripheral autoimmunity [178] Furthermore, in progesterone deficiency, a decrease in the number of Tregs (CD4+, CD25+, Foxp3, TGF-, and failure of lymphocyte adhesion and chemotaxis) has been linked to idiopathic infertility [179]. Recurrent implantation failures, recurrent miscarriages, pre-eclampsia, and intrauterine growth restriction are all associated with an insufficient number, impaired suppressive function, instability of Tregs, and inadequately regulated inflammation [180].

Ovulatory disorders and tonic high estrogen levels in the presence of a progesterone deficiency result not only in endometrial infertility but also endometrial hyperplasia due to chronically elevated estrogen levels that are not counteracted by progesterone [147]. Progesterone reaches the endometrial layer of the uterus during the second phase of the menstrual cycle and pregnancy, and exerts an immunosuppressive effect on natural killer (NK) cells, contributing to the correct implantation of the embryo and the further development of pregnancy [181]. Progesterone deficiency may negatively affect implantation disorders in women with PCOS also due to the lack of an immunosuppressive effect on the peripheral blood NK cells (PBNK), which express both isoforms of the classical progesterone receptor.

The immunomodulatory effect of progesterone is mediated by the progesterone-inducible blocking factor (PIBF) protein, which is produced in the presence of progesterone by progesterone receptor-positive lymphocytes. PIBF expression in lymphocytes shows an inverse correlation with NK cell activity. PIBF has been shown to increase the production of Th2 cytokines and inhibit NK cell degranulation [171]. In women with miscarriages and implantation disorders, lymphocytes do not produce this factor. The percentage of PIBF-positive lymphocytes in the peripheral blood of healthy pregnant women is much higher than in women at risk of premature birth [182].

Extremely important in reproductive medicine is the fact that progesterone, by binding to the glucocorticoid receptor (GR) in T lymphocytes, inhibits the production of pro-inflammatory IL-6 through dendritic cell-dependent pathways [178]. Therefore, in immune-related infertility, high doses of progesterone can retune the immune system and inhibit inflammatory responses that are normally blocked by steroid administration. The level of progesterone in pregnancy increases up to 10-fold in the serum, while in placental tissue it can be as much as 100-fold higher than in the maternal circulation. Thanks to this, it activates GR receptors and inhibits the differentiation of fetal T cells into Th17 cells associated with inflammation [179]. It has been proven that the GR receptor in T lymphocytes, stimulated by progesterone, mediates protection against autoimmunity in pregnancy [183].

Progesterone deficiency seems to be a key factor in the lack of control of autoimmune reactions triggered by estrogen and environmental factors, including specific autoantigens, including viral and bacterial infections. Progesterone, as a pro-pregnancy hormone, is necessary to sustain pregnancy which, in this context, is a semi-allograft. It triggers and maintains a specific, fetal-friendly environment in the endometrial cavity and in the mother’s system. On the one hand, it suppresses the rejection reactions of the fetus, whose antigens are half paternal and, therefore, foreign to the mother and, on the other hand, it triggers defense factors against infections to protect the newly developing body [176].

Recently, attention has also been paid to the important role of progesterone in immunomodulatory reactions and in inflammation of the fallopian tubes. Progesterone regulates the morphology and function of the epithelial cells in the fallopian tube, enabling a normal pregnancy. It reduces the contraction of the fallopian tube muscles and the frequency of the ciliary rhythm, which affects the transport of the oocyte and embryo. In addition, an increase in its level in the first stage of ovulation enables the correct capacitation of spermatozoa [184].

The multifaceted influence of various factors on the regulation of progesterone production, which contributes to the pathophysiology of such a complex syndrome as PCOS, is invariably important. It has been shown that the aforementioned adiponectin, the level of which is clearly reduced in PCOS patients, increases the secretion of progesterone and estradiol (E2), induced by IGF-1 in granulosa cells, by activating aromatase p450. Human ovarian granulosa cells have receptors for adiponectin. Thus, hypoadiponectinemia not only contributes to PCOS-related disorders by affecting IR and inflammation in the body, but it also reduces ovarian progesterone production [77,185].

## 10. Autoimmune Disorders in PCOS

Sexual dimorphism determines the directions of expression of cellular markers involved in innate and adaptive immunity. Women have better protection against infections, a predisposition to produce antibodies, and a predominance of the Th2 type response [186]. The higher expression of genes on the X chromosome includes the regulatory markers of the immune response, such as FoxP3 or CD40L. On the one hand, such immune system construction predisposes them to the possibility of accepting pregnancy in the context of a semi-allograft but, on the other hand, it exposes them to easier autoimmunity and auto-aggression [186]. Women of reproductive age are more likely to develop autoimmune diseases than men [187]. The natural consequence of these diseases is the presence of various types of auto-antibodies circulating in the woman’s body, which also affect oocyte maturation, embryonic implantation, or pregnancy development. Clinically, this is manifested by autoimmune infertility. Among PCOS patients, there is an imbalance of estrogen and progesterone, which results in an increase in autoantibodies. This does not always indicate a full-blown autoimmune syndrome, but the presence of antibodies is responsible for infertility [188,189]. Not only progesterone but also androgens and glucocorticoids are recognized as natural immunosuppressants. Estrogens, on the other hand, activate and strengthen humoral immunity. By affecting autoreactive B lymphocytes, they disrupt the mechanisms of immune tolerance and promote the secretion of autoantibodies [190].

Progesterone is a hormone that regulates key reproductive processes, from ovulation through implantation, placental development, and pregnancy. In addition, it has strong immunomodulatory properties that go beyond the reproductive system (Figure 4). In various tissues, it can act as a modulator of the inflammatory response [184]. The balance between the effects of estrogen and progesterone in the female body guarantees the proper course of immune reactions, including the control of autoimmunity [191,192]. Excessive activity of estrogen in relation to progesterone can lead to the formation of various autoantibodies and, consequently, affect a woman’s health [188]. It has been shown that the most active form of estrogen, 17-b estradiol, clearly increases the expression of growth markers and cell proliferation, for example, in macrophages and fibroblasts. Estrogens induce an autoimmune response and stimulate the production of pro-inflammatory cytokines such as IL-4, IL-1, IL-6, and interferon- by activating the complex NF-kB pathway [183].

In addition, an association has been shown between PCOS and autoimmune diseases such as systemic lupus erythematosus (SLE) and Hashimoto’s thyroiditis [188,193].

Antinuclear antibodies (ANA) in systemic diseases have a direct negative effect on fertility at several stages of the reproductive process. The presence of ANA in a woman’s system significantly interferes with the development of the oocyte and embryo and, thus, impairs the process of fertilization, implantation, and the course of pregnancy. In addition, the precipitation of immune complexes at the maternal-fetal interface and complement activation with inflammatory infiltration may also occur. All of these mechanisms can lead to miscarriage or premature birth [192]. Studies on SLE have shown that ANAs can also induce plasmacytoid dendritic cell activation via Toll-like receptor-9, resulting in increased production of inflammatory cytokines (such as IFN- α) [194]. These, in turn, stimulate the humoral immune response and lead to further ANA production. This “vicious circle” effect may be an important factor in inflammation in women with PCOS, in whom even a clinically insignificant presence of ANA may trigger fertility disorders [188]. Makled et al., showed significantly higher levels of ANA and anti-dsDNA among women with PCOS compared to 50 healthy women from the control group [195]. In contrast, another prospective study to assess serologic markers of autoimmunity in women with PCOS measured the serum levels of ANA and IgG autoantibodies against histones, nucleosomes, and double-stranded DNA (dsDNA) by enzyme immunoassays. In a group of 109 women with PCOS, significantly elevated serum levels of anti-histone and anti-dsDNA antibodies were observed compared to a control group of 109 healthy women of the same age. In contrast, serum ANA and anti-nucleosome antibody levels were similar in both groups. Interestingly, a significant correlation was found between ANA and TSH levels in the serum of women from both groups [196].

It turns out that in PCOS, the presence of many other autoantibodies has been documented in addition to ANA, antihistone, and anti-dsDNA,. Among them are: antisperm antibodies, anti-Sm antibodies (autoantibodies against the Smith antigen, components of the U1-snRNP complex), anti-carbonic anhydrase antibodies, anti-ovarian antibodies, and anti-pancreatic islet cell antibodies [188].

This association of reproductive failure with a propensity for various non-organ-specific autoantibodies has been shown not only in women with PCOS [179,197]. Elevated levels of autoantibodies in a group of women suffering from infertility may be the result of the manifestation of an autoimmune disease. This is supported by the diagnosis of abnormal autoantibodies in a group of clinically healthy women with numerous pregnancy losses, endometriosis, premature ovarian failure (POF), unexplained infertility, or failures of the IVF procedure [198].

The relationship between antibodies against pancreatic islet cells and the development of PCOS cannot be overlooked either. In one study, these antibodies were found in over 80% of PCOS patients [193]. The destruction of pancreatic beta cells results in a loss of insulin production and subsequent hyperglycemia. Hyperglycemia itself triggers autoimmune conditions, causes chronic inflammation of the endothelium, and may contribute to the development of PCOS. On the other hand, long-term glycemic disorders inevitably lead to a final lack of insulin. Treatment with insulin therapy can lead to weight gain, which will generate the already-described metabolic disorders. On the other hand, insulin itself will activate the ovary to overproduce androgens through IGF-1 and contribute to the development of PCOS. Such a sequence of events clearly shows the link between autoimmunity and metabolic disorders and the development of the clinical picture of PCOS. In this context, it is also important to remember latent autoimmune diabetes in adults (LADA), which is a subtype of autoimmune type 1 diabetes that begins in adulthood. As a result, PCOS patients with impaired carbohydrate metabolism may be misdiagnosed with pre-diabetes or DM2-related diabetes and receive ineffective treatment Such a situation will contribute to the lack of recovery of ovarian function, the persistence of inflammation, and infertility [199,200,201].

Other studies among women with PCOS showed a 3-fold higher incidence of autoimmune thyroiditis (AIT). All three types of anti-thyroid antibodies: anti-thyroid peroxidase antibodies (anti-TPO), anti-thyroglobulin antibodies (anti-TG), and anti-thyrotropic receptor antibodies (TRAbs)—are associated with reproductive failure [202]. The incidence of anti-TPO and anti-TG antibodies in PCOS patients was significantly higher compared to the control group [203]. Sarkar confirmed these data and showed a strong association of thyroid disorders with infertility and pregnancy loss among women. Both hyperthyroidism and hypothyroidism are detected in women who experience frequent miscarriages, fetal death, or delayed intellectual development in offspring [204]. In turn, the destruction of thyroid tissue causes oxidative stress, which leads to a disturbed balance in the body. It triggers a series of events leading to metabolic dysfunction, IR, and chronic inflammation in the endothelium of the whole body and in the ovarian tissue, which contributes to the development of PCOS. It turns out that progesterone plays a key protective role in this cascade of connections leading to autoimmune thyreiditis. It increases the differentiation of lymphocytes towards Tregs and their proliferation and, thus, directly counteracts autoimmunity. However, by binding the GR receptor on T lymphocytes, it inhibits the production of pro-inflammatory IL-6 by dendritic cells. Thus, chronic progesterone deficiency in PCOS may lead to autoimmunity, thyroid disorders, and infertility among such women [183,205,206]. Arduc et al., showed that estradiol was significantly higher in PCOS women with anti-TPO than in women with low levels of this antibody [207]. Moreover, anti-TPO correlated positively with the TSH, estradiol, and estradiol-to-progesterone (E/P) ratio.

The association of autoimmunity with PCOS is undoubtedly an important factor in the development of the syndrome itself as well as the cause of infertility among women with this syndrome. More and more areas are sought where the diagnosis of PCOS should increase vigilance as to the possibility of developing comorbidities. One of the recent reports describes the coexistence of non-infectious uveitis with the occurrence of pro-inflammatory factors in PCOS in women of reproductive age [208].

On the other hand, it is important to remember that autoimmunity itself can be a causal factor in PCOS. In autoimmunity, auto-reactive cells such as B and T lymphocytes are induced and antibodies are produced, which leads to the breakdown of the mechanisms responsible for self-tolerance [209]. Antibody-mediated mechanisms lead to the overproduction of free radicals, an increase in oxidative stress markers, and impairment of antioxidant mechanisms [210,211]. Such a sequence of events may lead to the inflammation of the ovarian tissue and trigger the pathomechanism leading to PCOS [209]. However, this does not mean that all women suffer from autoimmune diseases when diagnosed with PCOS. However, the susceptibility to immune disorders in this population is obvious and results from the mechanisms described above.

## 11. Summary

Infertility is associated with the suffering of individuals and affects the entire socio-demographic structure of the whole population. It results from immunological and metabolic abnormalities that initiate endocrine disorders. The main factors causing infertility among women with PCOS are impaired ovulation, implantation, and low-grade inflammation. On the one hand, IR and hyperandrogenism and, on the other hand, chronic inflammation, autoimmunity, and progesterone deficiency with a genetic and epigenetic background are the basis of fertility disorders and accompanying diseases in women with PCOS. Closely related immune-metabolic mechanisms underlie the development of this complex syndrome and cause infertility (Figure 4). Previous reports do not allow a clear conclusion to be drawn as to which of the above-mentioned mechanisms are primary in triggering PCOS and which only accompany them and, by strengthening their effect, contribute to the development of infertility in this syndrome. As a result, additional clinical studies should be conducted to better understand the relationships and implications of immunological, hormonal, and metabolic factors in order to develop new therapeutic strategies in PCOS patients to restore homeostasis and fertility.

## 12. Conclusions

Regardless of the presence of obesity and its obvious impact on the IR, a number of scientific studies suggest that women with PCOS exhibit an intrinsic form of IR that is unique to the disorder and contributes to infertility at the level of both ovulatory and implantation disorders.IR contributes to the development of the immunological processes that trigger inflammation, which has a negative impact on the processes taking place within the ovary. IR induces the overproduction of androgens in the ovaries through IGF-1 receptors, leading to hyperandrogenism and its aftermath.The influence of obesity on fertility may be associated with an increased leptin concentration, hypoadiponectinemia, a high concentration of triglycerides and free fatty acids in ovarian follicular fluid, and also the presence of oxidative stress and inflammatory mediators. This leads to abnormal ovulation and infertility.In recent reports, chronic, low-grade inflammation—regardless of the starting point—appears to be the most crucial factor contributing to the pathogenesis of PCOS and infertility.Oxidative stress present in the ovarian cells of PCOS patients has a negative impact on fertility, regardless of how the abnormal redox balance is triggered. It also points to a direct relationship between IR and immune system dysfunction by triggering oxidative stress and inflammation in the endothelium and, thus, also in the ovarian tissue.The effect of progesterone on the immune processes associated with ovulation, implantation, and pregnancy development confirms the key role of this immunomodulating hormone in reproduction. A chronic progesterone deficiency in women with PCOS, as well as hormonal and immunological imbalances, are unquestionably two of the most important causes of infertility in this syndrome.The balance between the effects of estrogen and progesterone in a woman’s body not only guarantees the proper hormonal balance that is conducive to maintaining fertility but also allows the inhibition of autoimmune reactions, also after supplementation in PCOS.The treatment of infertility in PCOS should be carried out by interdisciplinary teams of specialists so that patients are correctly diagnosed and the complex therapy successfully restores health and fertility.

## Figures and Tables

**Figure 1 biomedicines-11-01567-f001:**
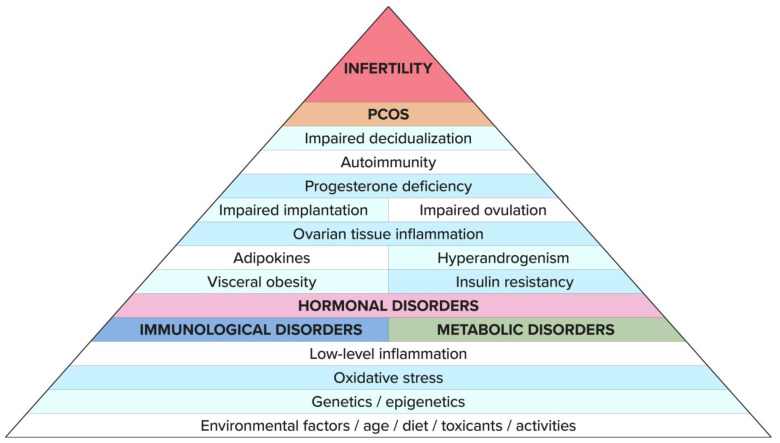
Infertility in PCOS—PYRAMID—“Tip of the Iceberg”. Infertility is the tip of the “iceberg” that is visible above the water’s surface. The part below the ocean’s surface results from generalized, low-grade inflammation, and disorders and diseases that—depending on the genetic, environmental, and cultural conditions related to lifestyle, diet, and exposure to harmful environmental factors—may manifest immediately or after many years from the infertility diagnosis.

**Figure 2 biomedicines-11-01567-f002:**
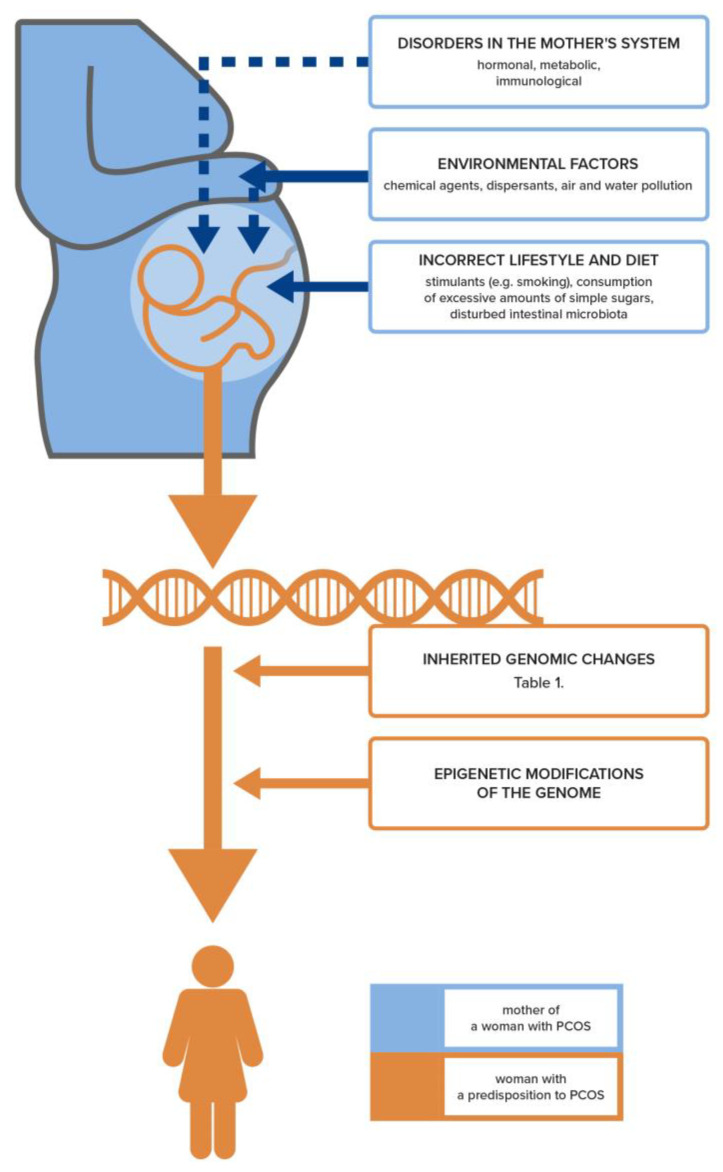
Genetic vulnerability to PCOS. There are genetic factors associated with the inheritance of PCOS. However, the mechanisms that trigger PCOS are due to epigenetic changes. These post-translational modifications of the genome depend on the environment of the maternal organism, which is increasingly subject to the influence of civilizational factors triggering metabolic, immunological, and, hormonal disorders involved in the PCOS pathomechanism of the developing fetus.

**Figure 3 biomedicines-11-01567-f003:**
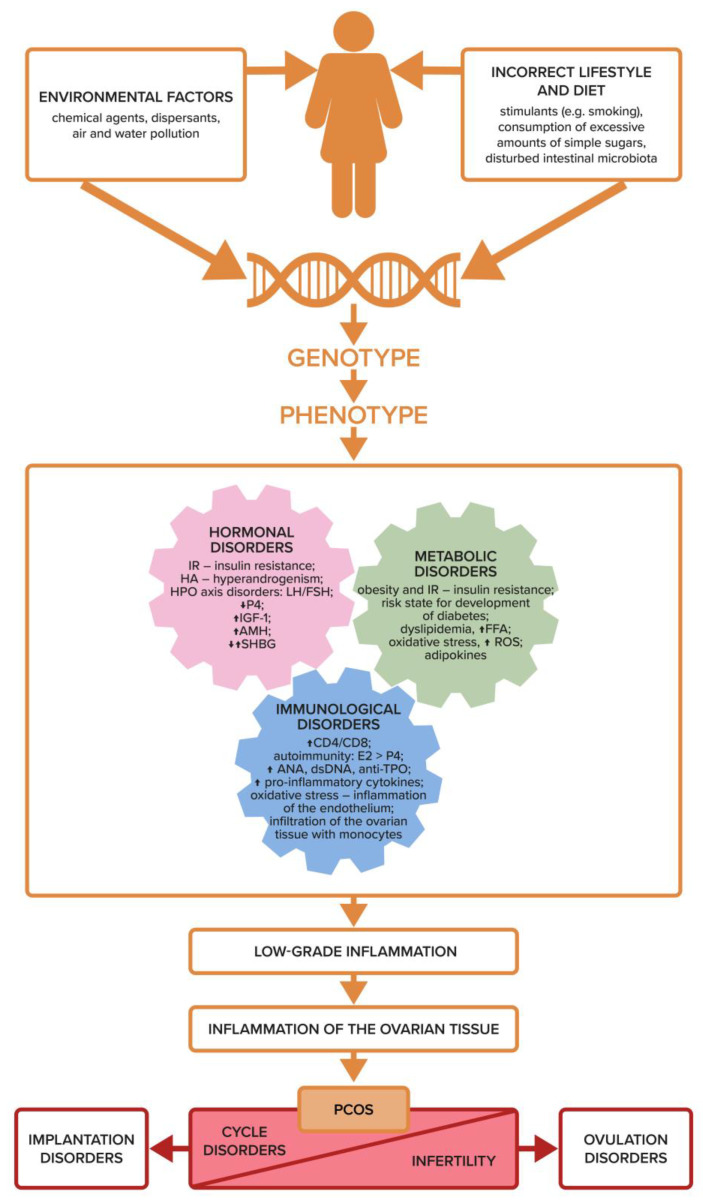
Triggers for infertility in PCOS. Depending on lifestyle, diet, and environmental factors, some genetic disorders appear earlier and others later. Several pathophysiological models are considered in PCOS, assuming it to be the primary disorder giving rise to this endocrinopathy and infertility. On the one hand, they include metabolic disorders such as IR, which can lead to immune disorders and oxidative stress. Insulin resistance (IR) is an important link in the pathogenesis of PCOS and is induced by inflammation/oxidative stress and hormonal dysfunction not only in adipose tissue (e.g., ovarian or adrenal androgens). It should be emphasized that even among thin women with PCOS, excessive consumption of glucose causes oxidative stress. Dysregulation of glucose metabolism and lipid metabolism causes low-grade inflammation in the endothelium, which also affects the ovarian tissue. On the other hand, the presence of chronic inflammation may contribute to the development of IR, which in turn is intensified by the release of androgens and pro-inflammatory cytokines from adipose tissue, not necessarily in obese women. Disorders caused by insulin resistance are sustained and further aggravated by the consequences of hormonal and metabolic dysfunctions, leading to immune dysregulation. IR also contributes to the dysregulation of the HPO axis, excessive secretion of androgens, and the lack/reduced production of adequate amounts of progesterone resulting from ovulation disorders. Progesterone deficiency promotes autoimmune processes that enhance IR and directly affect the ovulation process and oocyte quality and implantation. All this generates a cog-wheel effect, leading to systemic disorders in PCOS. The end result of this self-perpetuating mechanism is the abnormal growth of the ovarian follicles and impaired oocyte maturation, accompanied by endometrial receptivity dysfunction, which leads to irregularities of the menstrual cycle and infertility at various levels.

**Figure 4 biomedicines-11-01567-f004:**
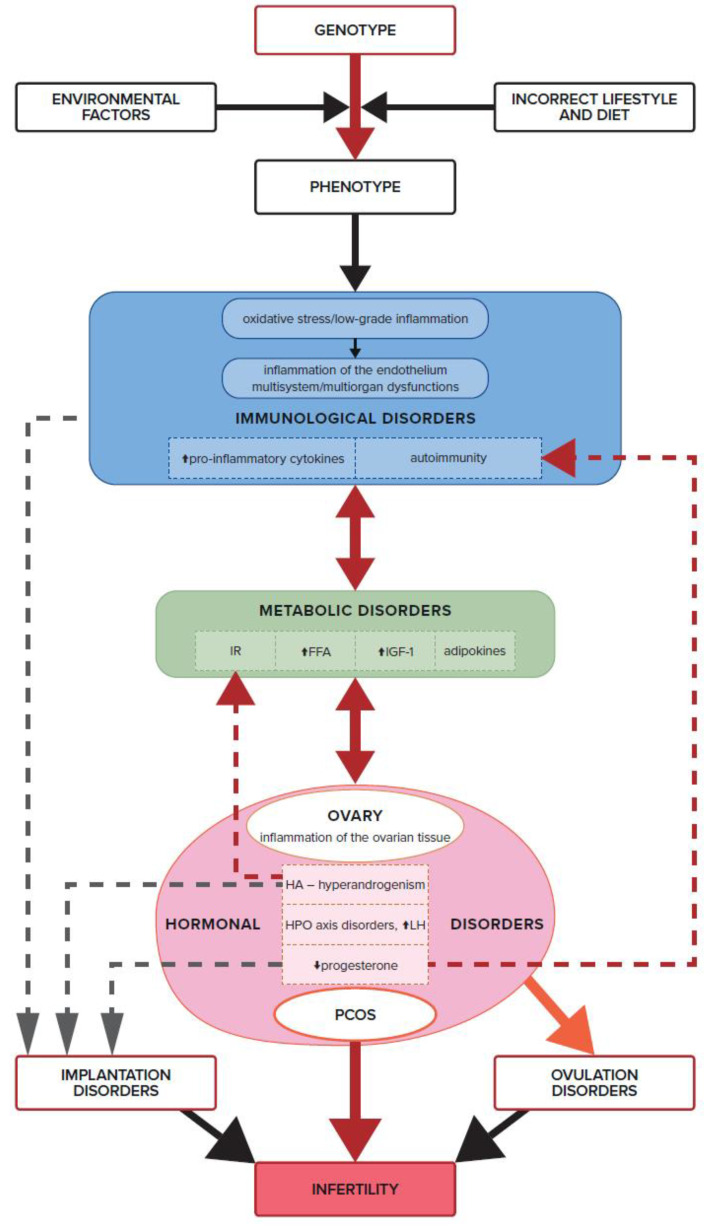
Summary: The basis of the described changes has its origin in genes and epigenetics. Immunological and metabolic disorders trigger an unfavorable hormonal profile. These dependencies interpenetrate and trigger each other and lead to a clinical picture of PCOS and infertility. This has been shown by red arrows, which most likely indicate the main pathways of the pathomechanism of PCOS and infertility.

**Table 1 biomedicines-11-01567-t001:** The main genes involved in pathophysiology of PCOS and their metabolic and fertility impact.

Gens Involves in the Pathophysiology of PCOS				
Steroidogenesis	Insulin Secretion and Action	Effect of Steroid Hormones	Gonadotropin Regulation	Others
*CYP21*, *CYP11a*, *CYP19*, *CYP17*	*IRS* group, *INSR*, *CAPN10*, *FTO*	AR, SHBG, *DENND1A*	FSHR, *LHCGR*, AMH, *HOXA* group, BMP	PAI-1
Hyperandrogenism	Diabetes, obesity Oxidative stress	Hyperandrogenism	Infertility	Infertility
**Ovulation and Implantation Disorders**				
**Infertility/Cycle Disorders**				

*CYP* group: cytochrome family p450; *INS*: insulin gene; *IRSR*: insuline receptor substrate; *CAPN10*: caplain—10; *FTO*: fat mass obesity; AR: androgen receptor; SHBG: sex hormone binding globulin; *DENND1A*: connecdenn-1; FSHR: follicle-stimulating hormone receptor; *LHCGR*: lutein hormone gen receptor; AMH: anti-Mullerian hormone; *HOXA*: gen responsible for succssesful implantation; BMP: bone morphogenetic protein; PAI-1: plasminogen activator inhibitor 1.

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
