# Peer review of "Immunological and Metabolic Causes of Infertility in Polycystic Ovary Syndrome"

_biomedicines, 2023, doi:10.3390/biomedicines11061567_

Round 1

Reviewer 1 Report

This is a comprehensive and nice review about the immunological and metabolic causes of infertility in PCOS. This review is organized in good shape and well written. However, the following concerns need to be clarified.

1. For figure 1, the author should clearly demonstrate the relationship between the left part (blue) and right part (gray). I am quite confused about the current figure 1.

2. I recommend the author to list all the factors in a table, with corresponding references combined, which will make it more clear for the readers to get all the information in a direct and easy way.

3. I suggest the author add several flowcharts to illustrate how these factors leads to the PCOS syndrome.

4. For the genetic factors which leads to PCOS, I recommend the author list a table of all the reported genes associated with PCOS.

The  review is organized in good shape and well written in English. 

Reviewer 2 Report

The authors made a huge effort to attempt to identify any immunological and endocrinological markers and even plausible causes for PCOS. This review appears comprehensive on the (1) endocrinological aspects alone and (2) immunological factors alone and association with PCOS.

However, this needs to be carefully considered as I do not see the direct links between obesity, adipokines, immunological factors as Causative for PCOS. Again, in young women, we do not diagnose PCOS until she is at least 8 years past her menarche. Also, obesity, IR and autoimmunity are not required to diagnose PCOS. Therefore, it is important that links and association should not be misinterpreted as causative for PCOS in adult women. In addition, not ALL PCOS women are obese or have IR or suffering from autoimmune conditions.

PCOS still has a lot to study, therefore, I find that all factors listed are good to group and categorize into a table format for readers to understand better. Additionally, the authors can create a figure that help the reader link the concepts they are proposing although I would caution the endocrinological and immunological factors as causative for PCOS. 

I would request for all the authors to offer more clarity and qualify that the published data reviewed are more associations and not causative.

English language is relatively accurate with some sentences structures needing further revision to ensure clarity and ease of understanding.

Round 2

Reviewer 1 Report

Overall the points that I raised have been addressed properly. I agree to publish this paper at the present form.

Author Response

We would like to thank you very much for the positive and motivating review of our manuscript entitled ‘Immunological and metabolic causes of infertility in polycystic ovary syndrome’ for publication in Biomedicines.

Please find enclosed a revised version of the manuscript.

We hope that the revised version will be suitable for publication in the Biomedicines.

Yours sincerely,

Anna Babińska

Reviewer 2 Report

The revised version is much improved with now with a number of figures to allow understanding of the detailed discussions between inflammation and adiposity. The table and figures summarized the very detailed explaining of the paper.

NA

Author Response

We would like to thank you very much for a positive motivating review of our manuscript

entitled ‘Immunological and metabolic causes of infertility in polycystic ovary syndrome’ for publication in Biomedicines.

The manuscript has been double-checked by a native speaker. Minor corrections have been made (in red).

Please find enclosed a revised version of the manuscript.

We hope that the revised version will be suitable for publication in the Biomedicines.

Yours sincerely,

Anna Babińska

Round 3

Reviewer 2 Report

Thank you for the revisions and attempts to summarize the thoughts linking up immunological, adiposity and ovulatory issues in PCOS - its a novel way to try to explore these potential mechanisms to determine the exact cause of PCOS to clinically manage this condition better. The figures 1 - 3 helped with the authors' hypotheses and train of thoughts however, figure 4 attempts to summarize the big points but it seems redundant as it still does not bring the lengthy discussion into more clarity. Therefore, I would suggest to re-do it or replace with a figure attempting to find common links or a flow chart of potential causal pathways which will make the huge paragraphs more palatable and summarize for the reader.

Author Response

We would like to thank you very much for a positive motivating review of our manuscript

entitled ‘Immunological and metabolic causes of infertility in polycystic ovary syndrome’ for publication in Biomedicines.

We have prepared new figure 4 to summarize the most important mechanisms of PCOS and their effects.

Please find enclosed a revised version of the manuscript.

We hope that the revised version will be suitable for publication in the Biomedicines.

Yours sincerely,

Anna Babińska